# Thiophene derivative inflicts cytotoxicity *via* an intrinsic apoptotic pathway on human acute lymphoblastic leukemia cells

Risa Mia Swain[1,2], Anahi Sanchez[1], Denisse A. Gutierrez[1], Armando Varela-Ramirez[1], Renato J. Aguilera[1]*

1 Department of Biological Sciences, The Border Biomedical Research Center, The University of Texas at El Paso, El Paso, Texas, United States of America, 2 Department of Molecular and Translational Medicine, Center of Emphasis in Cancer, Paul Foster School of Medicine, Texas Tech University Health Science Center El Paso, El Paso, Texas, United States of America

* raguilera@utep.edu

## Abstract

In an effort to identify novel anti-cancer agents, we employed a well-established High Throughput Screening (HTS) assay to assess the cytotoxic effect of compounds within the ChemBridge DIVERSet Library on a lymphoma cell line. This screen revealed a novel thiophene, F8 (methyl 5-[(dimethylamino)carbonyl]-4-methyl-2-[(3-phenyl-2-propynoyl) amino]-3-thiophenecarboxylate), that displays anti-cancer activity on lymphoma, leukemia, and other cancer cell lines. Thiophenes and thiophene derivatives have emerged as an important class of heterocyclic compounds that have displayed favorable drug characteristics. They have been previously reported to exhibit a broad spectrum of properties and varied uses in the field of medicine. In addition, they have proven to be effective drugs in various disease scenarios. They contain anti-inflammatory, anti-anxiety, anti-psychotic, anti-microbial, anti-fungal, estrogen receptor modulating, anti-mitotic, kinase inhibiting and anti-cancer activities, rendering compounds with a thiophene a subject of significant interest in the scientific community. Compound F8 consistently induced cell death at a low micromolar range on a small panel of cancer cell lines after a 48 h period. Further investigation revealed that F8 induced phosphatidylserine externalization, reactive oxygen species generation, mitochondrial depolarization, kinase inhibition, and induces apoptosis. These findings demonstrate that F8 has promising anti-cancer activity.

## Introduction

As cancer disparities increase as the years go by, it has become an urgent requirement for scientists to find alternative approaches to cancer treatment. It is projected that by the end of the year 2023, 59,610 new cases of leukemia will be diagnosed, and 23,710 people will die from the disease in the U.S. [1]. The overall leukemia incidence rate has increased by 1% annually in children since 1975 [1]. Cancer can be linked to evading normal apoptotic mechanisms and causing dysregulated proliferation of cancer cells [2]. Investigation of important biological pathways that play a role in apoptosis, may aid in understanding where the dysregulation

**Funding:** This research was funded by the National Institute of General Medical Sciences, under grant

number SC3GM088069-03, and the National Cancer Institute, under grant number 5R16GM149379-02, both awarded to RA. Core facilities were supported by the National Institute on Minority Health and Health Disparities (NIMHD) grant 3U54MD007592, awarded to RK. The funders had no involvement in the study design, data collection and analysis, decision to publish, or preparation of the manuscript.

**Competing interests:** The authors have declared that no competing interests exist.

occurs. New targeted chemotherapy agents are being explored and used to kill cancer cells specifically [3,4]. Many compounds in regular clinical use have displayed versatile applicability, some of which include anti-cancer activity.

After screening 1300 compounds, a novel thiophene compound was found to elicit potent cytotoxicity on the Acute Lymphoblastic Leukemia (ALL) CCRF-CEM cell line using the Differential Nuclear Staining (DNS) assay [9]. Thiophene is a heteroaromatic compound comprising of a five membered ring, with a sulfur atom located at the first position. Thiophene and its derivatives, with various substituents, constitute a significant group of heterocyclic compounds that are greatly important in medical chemistry, showcasing a range of intriguing applications [5]. Compounds containing a thiophene have exhibited diverse therapeutic properties, including anti-inflammatory, anti-psychotic, anti-anxiety, kinase inhibiting, and most importantly anti-cancer effects [5]. In the U.S. and Canada, thiophenes and their derivatives have demonstrated efficacy in treating diverse cancers and other clinical diseases [6,7]. For example, 2- Butylthiophene, a well-known thiophene has been used in the process of synthesizing anti-cancer agents. A serotonin agonist used in the treatment of Alzheimer's is derived from a maleate salt containing a thiophene, 1-(2,5-dimethylthiophen-3-yl)-3-(5-methyl-1H-imidazol-4-yl)propan-1-one [5]. These versatile compounds show promising activity, and here we will explore thiophene carboxylate F8 (Fig 1) anti-cancer activity *in vitro*. This study revealed that compound F8 exhibited cytotoxic effects on various human cancer cell lines. We examined this compound on a relatively sensitive cells, the Acute Lymphoblastic Lymphoma, CCRF-CEM cell line, to determine the likely cause of cytotoxicity and the mode of action this novel compound uses to induce cell death.

## Materials and methods

### Cell lines and culture conditions

F8 cytotoxicity was evaluated on seven human cancer and one non-cancerous human cell line. All utilized cell lines used in this study were acquired from the American Type Culture Collection (ATCC, Manassas, VA, USA).

**Fig 1. Chemical structure of F8 (methyl 5-[(dimethylamino)carbonyl]-4-methyl-2-[(3-phenyl-2-propynoyl) amino]-3-thiophenecarboxylate).**

Cell lines of blood cancer origin: CCRF-CEM (Acute Lymphoblastic Leukemia, ATCC CRL-2265), Jurkat (Acute Lymphoblastic Leukemia, ATCC CRL-2899), and NALM6 (Acute Lymphoblastic Leukemia, ATCC CRL-3273) were cultured with complete culture medium using RPMI-1640 (Hyclone, Logan UT, USA), with 10% fetal bovine serum (FBS, Hyclone), 100 μg/mL penicillin, and 100 μg/mL streptomycin (Thermo Fisher Scientific Inc. Rockford, IL). Of the cell lines of blood cancer origin, HL-60 (promyeloblast, ATCC CCL-240) required a 20% FBS in their culture media. In addition, a triple-negative breast adenocarcinoma cell line MDA-MB-231 (ATCC CRM-HTB-26), PANC-1 (ATCC CRL-1469), and A-375 (ATCC CRL-1619) were grown in Dulbecco's Modified Eagle's Medium (DMEM, CORNING, Corning, NY). DMEM medium was supplemented with 10% FBS and 100 U/mL penicillin, and 100 μg/mL streptomycin. Non-tumorigenic foreskin fibroblast cells, Hs-27 (ATCC CR-1634), were also cultured in a complete DMEM medium. Consistently, all cells were cultured in a 5% $CO_2$ humidified 37˚C incubator.

Prior to the use of the cell lines for the following experiments, the viability was assessed *via* flow cytometry using propidium iodide (P.I.) [8]. Cell populations with a 95% viability or higher were seeded into a multi-well plate.

## Preparation of compound

The experimental compound was sourced from the ChemBridge DIVERset library (ChemBridge Corporation, San Diego, CA) and was pre-diluted to 10 mM in dimethyl sulfoxide (DMSO). The compound (methyl 5-[(dimethylamino)carbonyl]-4-methyl-2-[(3-phenyl-2-propynoyl) amino]-3-thiophenecarboxylate), also known as F8, was found in said library set and used for the experiments detailed below. Aliquots and other concentrations of F8 were thawed and diluted with DMSO. The positive control, $H_2O_2$, was diluted in 1 x phosphate-buffered saline (PBS).

## Compound screening analysis

Thirteen hundred compounds were screened from the ChemBridge DIVERset library using the CCRF-CEM cell line. The primary screening was at a concentration of 10 μM by differential nuclear staining assay (Z' = 0.86) [9]. Upon screening, compounds that induced >50% cell death were subjected to a secondary screening to determine the 50% of cytotoxic concentration ($CC_{50}$) value. In the secondary screening, we used concentrations ranging from 0.01 μM to 30 μM to determine the concentration of the experimental compound required to disrupt 50% of the cell population [10].

## Differential nuclear staining to determine $CC_{50}$

The cytotoxicity compound F8 was assessed on the CCRF-CEM cell line through a DNS live-cell bioimaging assay [9]. CCRF-CEM cells were seeded in 96-well plates at a density of 10,000 cells per well, each containing 100 μl of tissue culture medium. Subsequently, the cells were then treated with a gradient of concentrations (0.01 μM to 5 μM) with compound F8 for a 48 h time point. The cells were plated with the compound itself, a solvent control (dimethyl sulfide; 1% DMSO; Millipore Sigma, St. Louis, MO USA), a positive control 1 mM hydrogen peroxide ($H_20_2$; Millipore Sigma, St. Louis, MO USA), and untreated cells for reference. To distinguish between both live and dead cells, each well received propidium iodide (P.I.; Biotium; Fremont, CA, USA) and Hoescht 33,342 (Invitrogen; Eugene, OR, USA) at a concentration of 5 μg/ml for each dye, with a 2 h incubation at 37˚C. Hoechst is a fluorescent dye that permeates both live and dead cells, whereas P.I. selectivity can only permeate cells whose plasma membrane is compromised. Images were captured from each well using a 96 multi-well plate reader ImageXpress Pico automated cell imaging system (Molecular Devices; San Jose, CA). The cytotoxic

concentrations were determined as previously described [9] using a linear interpolator calculator available at (https://www.johndcook.com/interpolator.html).

## Phosphatidylserine externalization

One of the well-known first indicators of apoptosis is the translocation of phosphatidylserine (PS) on the cell surface [11,12]. We sought to determine the mechanism of death that compound F8 induces in CCRF-CEM cells. Flow cytometry provides a convenient method for detection of phosphatidylserine (PS) through the conjugation of Annexin V-FITC, a member of a protein family that binds to the negatively charged phospholipids, serving as a marker for cells when PS translocates to the cell surface. In this study, CCRF-CEM cells were initially seeded in a 24-well plate at 100,000 cell density, each well containing1 mL of media. The cells were then incubated overnight with F8 $CC_{50}$ (obtained after 24 h incubation) and 2X $CC_{50}$ (2.89 µM and 5.78 µM). For the controls, DMSO was used for the vehicle control; as a positive control, cells were treated with 1 mM $H_2O_2$, and untreated cells were used to evaluate the baseline of the dead cell population, acting as a negative control for cytotoxicity.

Following treatment, the cells were then collected and placed on flow tubes on ice. They were then washed with ice-cold PBS and centrifuged at 1200 RPM for 5 min. The resulting pellets were then resuspended in Annexin V-FITC and P.I. in 100 µl of binding buffer and incubated in the dark on ice for 15 min. Ice-cold binding buffer (400 µl) was then added before the analyses and read immediately by the flow cytometer. Approximately 10,000 events/cells were acquired per sample by using the Kaluza software (Backman Coulter), and samples were done in triplicates. The total apoptotic cell populations were determined by combining both the early and late stages of apoptosis [19]. We conducted three separate measurements and calculated the average and standard deviation, resulting in the determination of the percentages of both apoptotic and necrotic cells.

## Assessing mitochondrial membrane potential

To assess the integrity of the mitochondrial membrane in CCRF-CEM cells following exposure to F8, we employed the MitoProbe JC-1 Assay Kit from Molecular Probes (M34152). The fluorescent cationic probe (5′,6,6′-tetrachloro-1,1′,3,3′-tetraethylbenzimidazolylcarbocyanine iodide JC-1) is able accumulate inside the mitochondria. In mitochondria where the membrane is intact, the JC-1 dye will aggregate and emit red fluorescent complexes. In mitochondria with a damaged membrane, the JC-1 dye cannot accumulate and form aggregates, staying as monomers that emit green fluorescence [13,14]. This red/green fluorescence serves as a dependable indicator of mitochondrial integrity. In this assay, CCRF-CEM cells were first seeded in a 24-well plate in 1mL complete culture media at a density of 100,000 cells per well. The cells were then subjected to treatment with F8 24 h $CC_{50}$ and 2X $CC_{50}$ (2.89 µM and 5.78 µM). The controls for this assay are the same as previously described, and each sample was run in three independent measurements. Following incubation, the cells were gathered, centrifuged at 1200 RPM, and resuspended in 500 µl of PBS containing 2 µM of JC-1 dye. Following this, the cells were then incubated at 37˚C for 30 min and then washed with warm PBS prior to analyses. When examined in the flow cytometer, cells emitting only a green fluorescence signal (~529 nm) indicate mitochondrial depolarization (damage). In contrast, cells emitting a red fluorescent signal (~590 nm) indicate that their mitochondria are polarized and in a healthy state.

## Reactive oxygen species accumulation

Reactive oxygen species (ROS) in the cell produce oxidative stress in the mitochondria by retrograde redox signaling [15]. The generation of reactive oxygen species can serve as an

indicator of mitochondrial dysfunction in the intrinsic apoptotic pathway [16]. To assess the levels of ROS in CCRF-CEM cells, we utilized the 6-carboxy-2′,7′-dichlorodihydrofluorescein diacetate (carboxy-H2DCFDA) reagent (Invitrogen, C400) [15]. Intracellular esterase's cleave this nonfluorescent carboxy-H2DCFDA leading to oxidation and forming a green fluorescent product 2′,7′-dichlorofluorescein (DCF) [17]. The cells were seeded in a 24-well plate in 1 mL of complete media, at a density of 100,000 cells per well and treated with F8 24 h $CC_{50}$ and 2X $CC_{50}$ (2.89 μM and 5.78 μM) with the same controls as previously mentioned for an 18 h incubation. As previously mentioned, the experimental and control treatments were evaluated in three independent measurements and normalized. Post incubation, the samples were collected and centrifuged at 1200 RPM for 5 min. Afterward, they were resuspended in a pre-warmed mixture of carboxy-H2DCFDA and 1 x PBS, making a final concentration of 10 μM dye. Cells were then incubated for 1 h at 37˚C, followed by another centrifugation for 1200 RPM for 5 min. Following resuspension in 500 μl PBS, the cells were given a 30 min recovery period and then analyzed by the flow cytometer.

## Caspase 3/7 activation

Upon initiation of apoptosis, caspase 3/7 are activated to proceed with the process of apoptosis. The activation of caspase 3/7 within the cells was evaluated by using a cell membrane permeable fluorogenic NucView 488 caspase substrate [18]. By permeating living cells with an intact membrane, the substrate is able to be cleaved. This process forms a high-affinity DNA dye that stains the nucleus green [19]. CCRF-CEM cells were treated with F8 24 h $CC_{50}$ and 2X $CC_{50}$ (2.89 μM and 5.78 μM) for 8 h. The cells were harvested and centrifuged at 1200 RPM for 5 min. The pellets were resuspended in 200 μl of PBS containing 5 μl of NucView488 caspase 3/7 substrate. This mixture was then incubated in the dark at room temperature for 45 min. Following the incubation, 300 μl of PBS was added to each flow tube and read immediately *via* a flow cytometer. In the flow cytometer, a green fluorescent signal indicates that caspase 3/7 activation is occurring within the cell. Each sample was performed in three independent measurements utilizing the Kaluza software (Beckman Coulter).

## Cell cycle analysis

To investigate F8 ability to induce alteration in cell cycle profile, CCRF-CEM cells were plated in 24-well plates at a density of 100,000 cells per well, with each well containing1 ml of media [20]. CCRF-CEM cells were treated with F8 $CC_{25}$ and $CC_{10}$ (1.44 μM and 0.57 μM obtained after 24 h) for 72 h. The controls that were used for this assay are the same as described above. After the 72 h incubation, the cells were harvested in flow cytometry tubes and centrifuged at 1200 RPM for 5 min. The cells were then resuspended in 100 μl of room temperature media. The nuclear isolation medium (NIM)-DAPI reagent was resuspended in each flow tube (200 μl) and analyzed *via* flow cytometry. The NIM-DAPI dye can stain the cell nuclei, aiding in quantifying the total amount of DNA per cell. As previously stated, the samples were run in three independent measurements, and 10,000 events per sample were acquired (Gallios; Kaluza software; Beckman Coulter).

## MAPK and JAK-STAT array

To gage the phosphorylation of essential biological pathways sometimes responsible for oncogenesis, a MAPK and JAK-STAT Array was conducted (RayBiotech Human/Mouse MAPK Phosphorylation Array AAH-MAPK-1-2, Human JAK-STAT Pathway Phosphorylation Array C1 AAH-JAK-STAT-1-2). This experiment utilized the CCRF-CEM cell line. The cells were treated with 2X $CC_{50}$ (5.78 μM) of F8 and DMSO as vehicle control for 4 h. The cells were

then harvested in 15 ml conical tubes, pelleted, and transferred to 1.5 ml tubes. The cells were then washed with 1 ml of PBS and pelleted again by centrifugation at 1000 g for 6 min. Following the centrifugation, the PBS was removed, and the pellet was resuspended in the 1X working solution of the Cell Lysis Buffer containing Protease Inhibitor Cocktail and Phosphatase Inhibitor Cocktail. The cells were rocked gently at 4˚C for 1 h. The total protein extracts were then transferred to microcentrifuge tubes and centrifuged for 1200 RPM for 10 min. The protein concentrations were determined utilizing a bicinchoninic acid (BCA) assay for total protein concentration (Thermo Scientific™ Pierce™ Rapid Gold BCA Protein Assay Kit, A53225). Protein lysates were stored for later use.

Membranes for the MAPK and JAK-STAT were blocked using 2 ml of blocking buffer for 30 min at room temperature. The blocking buffer was then removed, and 1 ml of the undiluted sample was put into each well with the membranes and incubated overnight at 4˚C. Following the overnight incubation, the samples were aspirated from each well and washed with washer buffer I for 5 min at room temperature; this wash was repeated 3 times. The membranes were washed with washer buffer II for 5 min at room temperature; this step was performed 2 times. One ml of the prepared detection antibody cocktail was pipetted into each well and incubated overnight at 4˚C. The membrane was then washed in the same manner as previously mentioned, and 2 ml of 1X HRP-Anti-Rabbit IgG was pipetted into each well and incubated for 2 hours. The HRP-Anti-Rabbit IgG was pipetted from each well and washed in the same manner. The membranes were then transferred to blotting paper to remove any excess washer buffer and placed printed side up onto a plastic sheet lying on a flat surface. The detection buffer mixture was added onto each membrane and incubated for 2 min at room temperature. Another plastic sheet was placed on top of the membranes and gently rolled across in order to smooth out air bubbles. The sandwich membranes were then read *via* the Thermo Fisher iBright for chemiluminescence imaging. Densitometry was obtained using the Thermo Fisher iBright Densitometry analysis software. The data was then normalized per the manufacturer's recommendation.

## Statistical analyses

Each data point within the experiment was derived from a minimum of three separate measurements. A total of eight independent measurements were utilized to determine the overall cell count in the samples treated with the vehicle in the conducted DNS assays. To calculate the 50% cytotoxic concentration (CC50), we utilized linear interpolation as previously described. Statistical significance was established by computing P-values using a two-tailed paired Student's t-test to compare the vehicle-treated and experimental groups, using the t-test or the 2 independent Means software accessible at (https://www.socscistatistics.com/tests/studenttest/default2.aspx). Significant P-values, indicated by asterisks, are as follows: * P<0.05, ** P<0.01, and *** P<0.001.

## Results

### Detection of cell death *via* the differential nuclear staining (DNS) assay

The cytotoxic effect of F8 was analyzed using the Differential Nuclear Staining imaging assay on seven human cancer cell lines and one non-cancerous fibroblast cell line. The DNS assay, which has a robust Z' factor of 0.86, was utilized because it is reliable and validated for primary and secondary screenings for potential cytotoxic compounds [9,10]. Hoechst fluorescent dye is able to permeate both living and dead cells when exposed for 2 h before reading, providing the total number of cells [21]. In contrast, P.I. selectively permeates cells whose membrane has been compromised, thus labeling only dead cells. As indicated in Table 1, F8 displays a

**Table 1. Cytotoxicity values of F8.**

| Cell Type | Cell Line | CC$_{50}$ μM | Mean ± S.D. | SCI* |
|---|---|---|---|---|
| Acute Lymphoblastic Lymphoma | CCRF-CEM | 0.856 | 0.356 | 34.35 |
| Acute T Cell Leukemia | JURKAT | 0.805 | 0.021 | 36.53 |
| Acute Promyelocytic Leukemia | HL-60 | 1.29 | 0.356 | 22.79 |
| Pancreatic Carcinoma | PANC-1 | 1.48 | 0.07 | 19.87 |
| Malignant Melanoma | A375 | 2.69 | 0.40 | 10.93 |
| Breast Adenocarcinoma | MDA-MB-231 | 3.05 | 0.32 | 9.64 |
| Normal Foreskin Epithelial | Hs-27 | 29.41 | 0.127 | - |

Determination of the 50% cytotoxic concentration (CC$_{50}$) and Selective Cytotoxic Index (SCI) on multiple cancer cell lines at a 48 h time point.

*SCI index values were calculated using the following equation: CC$_{50}$ of non-cancerous cells (Hs-27 fibroblasts) divided by the CC$_{50}$ of the cancer cell line.

gradient of cytotoxicity across various human cancer cells tested. All grown cell lines were incubated at 48 h with F8 to obtain their CC$_{50}$ value. The adherent cell lines, A373, PANC-1, MDA-MB-231, and Hs-27, were initially left to incubate overnight to allow their attachment to the bottom of the plate in media. This is done solely in the absence of the compound so the cells can properly adhere. Lastly, a concentration gradient of F8 in triplicate was added to the cells. After an extra 48 h incubation, a dose-response curve was generated for calculating the concentration at which 50% (CC$_{50}$) of the cell population have died. The values depicted in the Table 1 range from 0.856 μM on CEM to 29.41 μM on the Hs-27 cell line. The data suggest potential selectivity in leukemia and lymphoma cell lines, with the CC$_{50}$ being at or below 1 μM to the adherent cell lines whose CC$_{50}$ is above 1 μM. In this study, we continued our experiments in the CCRF-CEM cell line to partially elucidate the mechanism of action of F8.

## F8-induced phosphatidylserine externalization in CCRF-CEM cells

To determine whether F8 causes death *via* apoptosis or necrosis, the detection of phosphatidyl-serine (P.S.) externalization was conducted in F8-treated CCRF-CEM cells. For this purpose, the Annexin V-FITC and propidium iodide (P.I.) assay and a flow cytometry protocol was utilized [9]. Moreover, we used the CC$_{50}$ and twice (2X) the CC$_{50}$ concentrations that were determined on CCRF-CEM. The data revealed that there is significant P.S. externalization compared to the DMSO-treated (solvent control) CCRF-CEM cells. As expected, there was a high percentage of P.S. externalization in the cells treated with the positive control H$_2$O$_2$, whereas the cells that were untreated or treated with the vehicle showed low P.S. externalization (Fig 2). The results indicate that F8 initiated apoptosis to signal for cell death due to the significant P.S. externalization in CCRF-CEM cells.

## F8 caused mitochondrial membrane depolarization

High levels of ROS can depolarize the mitochondria and cause damage to DNA, proteins, and lipids [22,23]. Many conditions that induce oxidative stress on the mitochondria evoke apoptosis [23]. We evaluated the effect of F8 on CCRF-CEM to detect mitochondrial depolarization using the JC-1 reagent and flow cytometry. Cells were incubated for 4 h with CC$_{50}$ (obtained after 24 h of incubation), and two times (2X) the CC$_{50}$ (2.89 μM and 5.78 μM), and the integrity of the mitochondria was assessed. These analyses revealed the F8 induced significant depolarization to the cell's mitochondria at the CC$_{50}$ and 2X CC$_{50}$ concentration compared to the vehicle control (Fig 3). Thus, F8 was able to depolarize the mitochondria in CCRF-CEM cells.

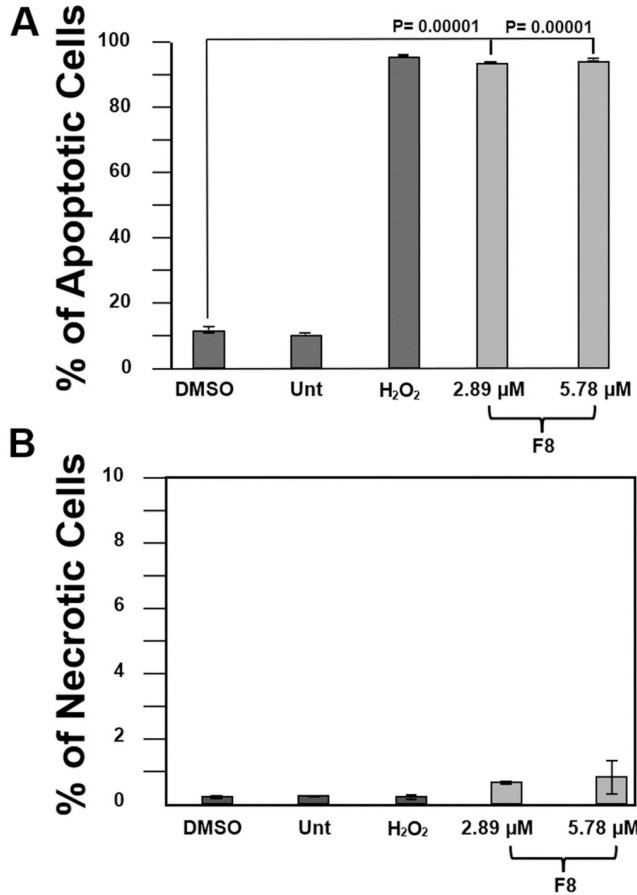

**Fig 2. F8's induction of apoptosis.** F8 compound-induced apoptosis in CEM cells. F8 ability to induce apoptosis was assessed in a phosphatidylserine externalization assay and measured through flow cytometry. Analysis was performed following a 24 h incubation with 24 h $CC_{50}$ and $2X$ $CC_{50}$ (2.89 μM and 5.78 μM). Controls included DMSO as the vehicle, $H_2O_2$ as the positive, and untreated. Significant phosphatidylserine externalization was evident for the $CC_{50}$ and $2X$ $CC_{50}$, given the p-value $p < 0.00001$ (***).

## Induction of ROS by F8 treatment

ROS generation indicates a disturbance in homeostasis, and their accumulation promotes apoptosis [24]. ROS accumulation was identified with carboxy-$H_2$DCFDA reagent and analyzed by flow cytometry. To examine the accumulation of ROS, we treated CCRF-CEM cells with F8 $CC_{50}$ and $2X$ $CC_{50}$ (2.89 μM and 5.78 μM) and incubated them for 18 h. Significant ROS accumulation in F8-treated cells was evident ($p = 0.0005$, $p = 0.003$; Fig 4), suggesting the contribution of oxygen radicals to apoptosis.

## Activation of executioner caspases 3/7

Caspases play a crucial role in both triggering and carrying out the process of apoptosis. Caspases can be categorized into three groups, initiator caspases (caspases 2, 8, 9, and 10), executioner caspases (caspases 3, 6, and 7), and lastly, inflammatory caspases (1, 4, 5, 11, and 12) [25]. Caspase activity begins upon activation, and this is followed by nucleic acid cleavage. To affirm that F8 induces apoptosis, the activation of caspase 3/7 was measured in CCRF-CEM cells by flow cytometry. The cells were treated with $CC_{50}$ and $2X$ $CC_{50}$ (2.89 μM and 5.78 μM)

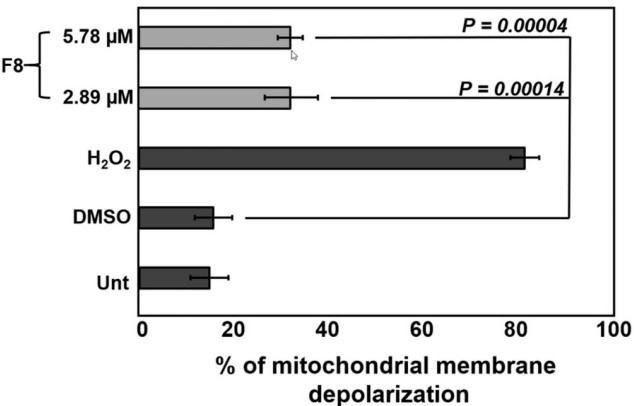

**Fig 3. Depolarization of the mitochondria.** Significant mitochondrial depolarization was induced by compound F8 on CEM cells following a 4 h incubation treated with the $CC_{50}$ and 2X $CC_{50}$ (2.89 μM and 5.78 μM). The cells were stained with JC-1 reagent and analyzed by flow cytometry. Statistical analyses were obtained through the two-tailed Students paired *t*-test. The controls that were included are the same as previously stated.

and incubated for 8 h. Increased caspase was observed at the $CC_{50}$ and 2X $CC_{50}$ concentration and compared to the DMSO control (Fig 5). The confirmation of Caspase 3/7 activation indicated that F8 initiated the intrinsic apoptotic pathway as its mechanism of action.

## F8 causes DNA fragmentation

To assess the effect F8 has on the cell cycle profile, the distribution of phases was evaluated on CCRF-CEM cells. The cells were treated with $CC_{25}$ and $CC_{10}$ (1.44 μM and 0.57 μM) for 72 h.

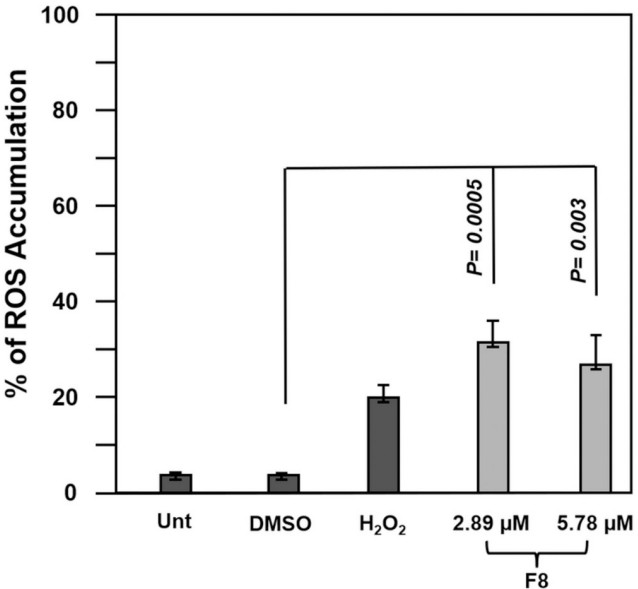

**Fig 4. F8 induced ROS accumulation.** Significant ROS was induced by F8 $CC_{50}$ and 2X $CC_{50}$ (2.89 μM and 5.78 μM) in CCRF-CEM cells compared to the vehicle control, DMSO, following an 18 h incubation period. Statistical analyses were acquired using a two-tailed t-test, p = 0.0005 and p = 0.003.

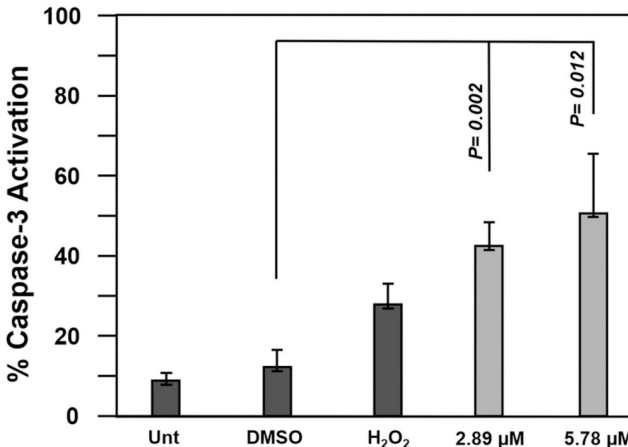

**Fig 5. F8 induces caspase activation.** Compound F8 $CC_{50}$ and 2X $CC_{50}$ (2.89 μM and 5.78 μM) induced significant caspase 3/7 activation in CEM cells. The cells were incubated for 8 h and stained with NucView 488 caspase-3/7 substrate. Statistical analyses were obtained using the two-tailed Student's paired t-test compared to the vehicle control (1% DMSO).

It is important to note that at the 24 h $CC_{50}$ concentration, extensive damage was observed and interfered with the cell cycle analyses, and for this reason, lower concentrations were used to observe cell cycle progression. The same controls were used in this experiment as those presented in the above sections. Although no significant changes in cell cycle progression were observed after F8 treatment, the induction of DNA fragmentation by F8 was demonstrated by an increase in the sub $G_0$-$G_1$ phase (Fig 6). The p-values of DMSO-treated cells compared with the experimental treatments were *p = 0.007* and *p = 0.027*, respectively. In conclusion, DNA fragmentation was evident upon the treatment of F8 on the CCRF-CEM cell line after 72 hours.

## F8-induced hypophosphorylation of critical biological enzymes

The phosphorylation of the MAPK and JAK-STAT were assessed on CEM cells with compound F8 with the use of the Human Phosphorylation Pathway Profiling Array C55. Cells were plated at a density of 2.7 x $10^6$ per mL and treated with F8 2X $CC_{50}$. Lysates of the vehicle (DMSO) and experimental samples were obtained and used for the membranes. Following exposure, the data was obtained *via* densitometry and normalized according to the protocol data analysis. As shown in Fig 7, signal reduction was observed for JAK-STAT from F8-treated cells compared to the DMSO-treated cells. The fold reduction for JAK1 was 0.74 in comparison to 1 for DMSO. The fold changes for JAK and STAT are as follows: JAK 2 (0.69), STAT 2 (0.64), STAT 3 (0.66), STAT 5 (0.69), STAT 6 (0.58), and TYK-2 (0.57). Interestingly, F8-treatment did not affect STAT 1 (0.86) and phosphorylation. The impact of JAK-STAT signaling on cancer cell survival, proliferation, invasion of tumors, coupled with the hyperactivation of STAT3 and STAT5; has positioned the JAK-STAT pathway as a potential pathway for cancer therapy [26]. Given the data, there was a reduction in the JAK and STAT pathways suggesting the compound may be inhibiting an essential biological pathway responsible for tumor progression. In addition to the JAK-STAT pathway, there was a reduction in the fold phosphorylation of other important regulators, most noticeably of the SHP Protein Tyrosine Phosphatase proteins SHP 1 (0.39) and SHP 2 (0.68) that would be predicted to reduce phosphatase activity. There was also a minor reduction in phosphorylation of the proto-oncogene SRC tyrosine-

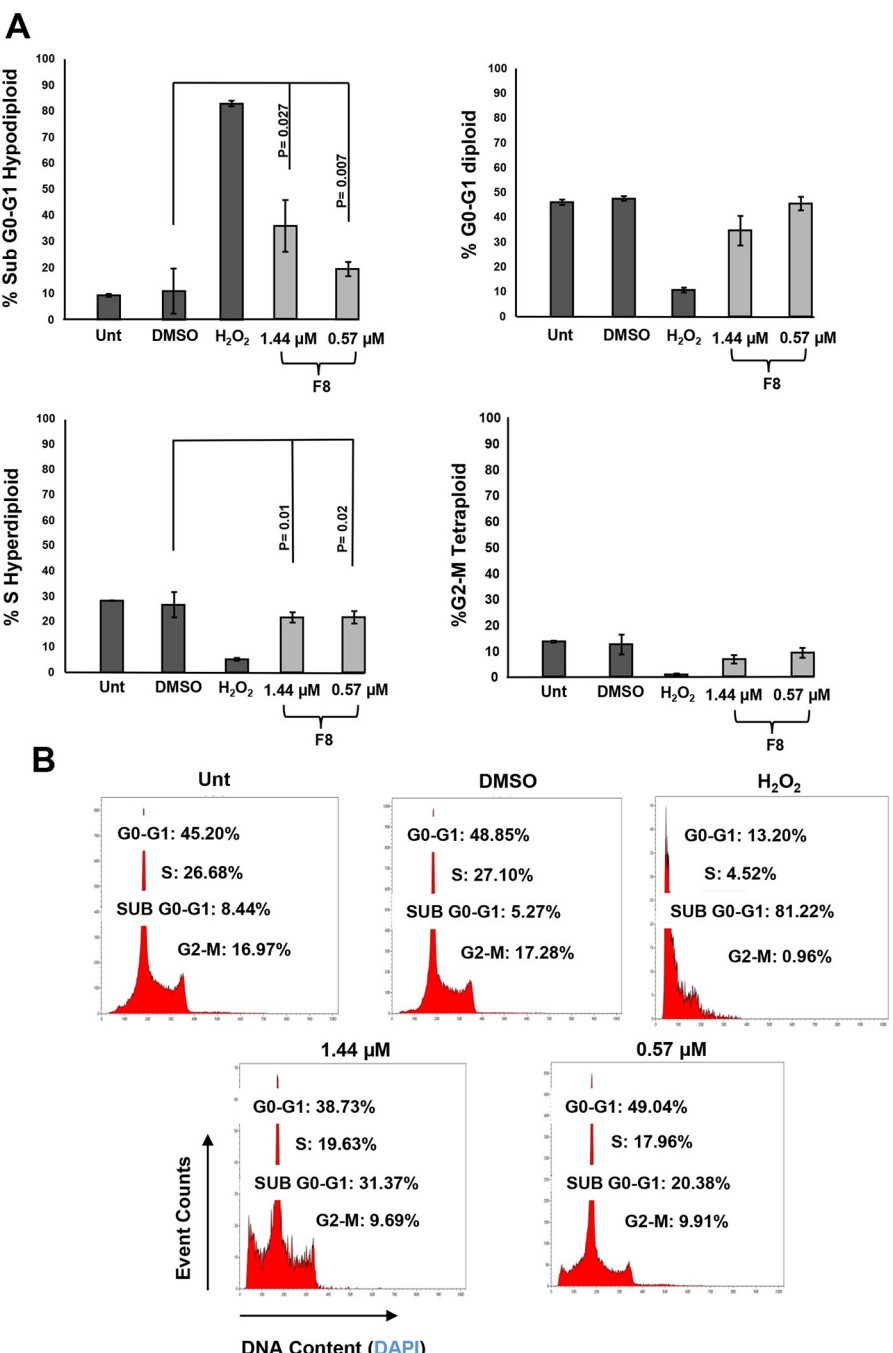

**Fig 6. F8 induced DNA fragmentation on CEM cells, a hallmark of apoptosis.** F8 cytotoxicity was analyzed *via* flow cytometer after a 72 h exposure on CEM cells utilizing $CC_{25}$ and $CC_{10}$ (1.44 μM and 0.57 μM). NIM-DAPI was used to stain each cell's DNA amount and quantified in the flow. F8 displays DNA fragmentation evident in the Sub G0-G1 phase. Controls were included, DMSO as the vehicle and $H_2O_2$ as the positive control, and untreated cells, respectively.

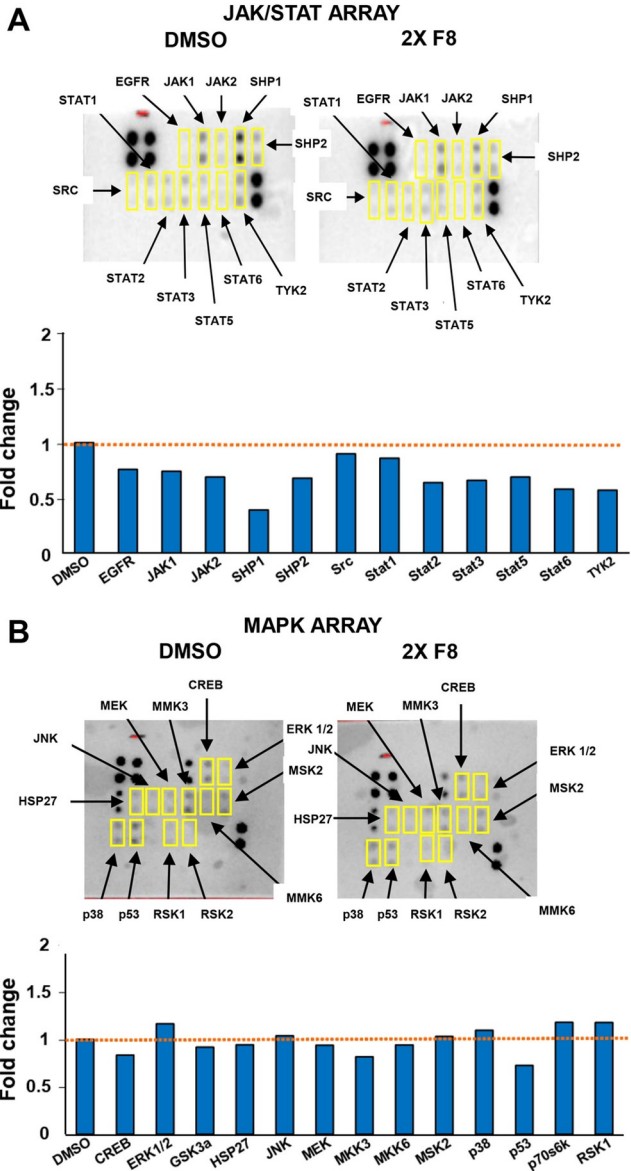

**Fig 7. F8 causes hypophosphorylation in JAK/STAT pathway.** Human Phosphorylation Kinase Array C55 was conducted on the CCRF-CEM cell line and treated with 2X CC$_{50}$ (5.78 μM). This assay was used to determine the phosphorylation status of MAPK and JAK-STAT membranes to gauge phosphorylation. Fold changes were determined using densitometry. Hyperphosphorylation was observed in the MAPK membrane versus the JAK-STAT. Hypophosphorylation was displayed in the JAK 1–2 and STAT 2–5.

protein kinase (0.9) and a slight decrease in the phosphorylation of Epidermal Growth Factor Receptor (EGFR 0.76).

In contrast, there were minor phosphorylation effects on MAP Kinase array proteins after F8-treatment, with values for ERK 1/2 (1.16), MSK2 (1.03), CREB (0.83), HSP 27 (0.94), JNK (1.04), MEK (0.94), MMK3 (0.82), MMK6 (0.94), P38 (1.10), P53 (0.73), and RSK 1 (1.18).

Our results indicate that F8 can downregulate members of the Jak/Stat pathway, suggesting a possible mechanism for induction of cell death.

## Discussion

Previously it has been shown that thiophene compounds can induce apoptosis in cancer cell lines [27]. In the U.S., thiophenes have been utilized for the treatment of various cancers and other medical conditions [28]. Thiophenes are a sulfur-containing heterocyclic compounds that bind with a wide range of cancer-specific protein targets, depending on its nature [28]. Here we discuss the effects of thiophenecarboxylate compound, F8, and its ability to induce cell death in multiple cancer cell lines. F8 was found to cause cytotoxicity with $CC_{50}$ values ranging from nanomolar to low micromolar concentrations (0.805 μM to 3.05 μM). Additionally, we demonstrated the compounds' ability to induce apoptosis in the CCRF-CEM cell line by investigating phosphatidylserine externalization, the generation of reactive oxygen species, assessing the mitochondria's membrane potential, and the activation of executioner caspase-3/7. Based on these assays, it was concluded that F8 induced cell death *via* the apoptotic pathway.

A cell in the process of apoptosis can follow through either the intrinsic or extrinsic pathway [29]. The intrinsic pathway involves the release of several factors *via* the mitochondria within the cell [30]. We performed a mitochondrial depolarization assay (JC-1) to assess the integrity of the mitochondria, an indication observed during apoptosis triggered through the intrinsic apoptotic pathway. Compound F8 induced significant depolarization of the mitochondria, thus proving F8 acts *via* the intrinsic apoptotic pathway.

We studied ROS generation in F8-treated CCRF-CEM cells. ROS generation is critical because it induces the depolarization of the mitochondria's membrane [31]. Excessive generation of ROS can induce cellular stress, activates caspases, and ultimately results in cell death [32]. As shown in Fig 4, F8 induced ROS overproduction in CEM cells, thus leading to apoptosis.

By permeabilization of the mitochondrial membrane, the apoptotic executioner caspase-3 becomes activated [30]. Typically, an apoptotic death is irreversible after caspase-3 is activated [33]. Upon activation, caspases demolish key proteins and activate other enzymes. Caspase-3/7 activation was assessed in CCRF-CEM following treatment with F8. Significant caspase-3/7 activation was shown thus confirming that apoptosis was activated *via* the intrinsic pathway as part of the machinery used by F8 to inflict cell dead.

A cell cycle analysis measured cellular DNA content in CCRF-CEM cells upon treatment with F8. Interestingly, treatment with F8 did not induce significant change to the cell cycle profile in the CCRF-CEM cell line but caused DNA fragmentation. Cells in the process of apoptosis demonstrate a distinct morphological trait, DNA fragmentation, thus confirming our outcomes. [34].

A phosphorylation (activation) array was performed using thiophene F8-treated CCRF-CEM cell samples. We sought to determine if F8 can inhibit phosphorylation in the MAPK, JAK, STAT, and ERK pathways since thiophenes have been previously shown to inhibit the phosphorylation of essential biological pathways [27]

Our results indicate that JAK 1 and JAK 2 were hypophosphorylated after F8 treatment. The JAK-STAT pathway, a significant route in cancer development, consolidates signals from diverse cytokines, hormones, and growth factors to either activate or suppress gene expression.[35]. It's been shown that the persistent activation of JAK 2 and STAT 3 found in many different cancers play essential roles in the development of malignancies [26]. Hyperactivation of JAK 2 promotes tumorigenesis, the over proliferation of tumors, and the overall survival of cancer cells [26]. In certain instances, the continuous activation of JAK 2 has been implicated in the survival, proliferation, angiogenesis, evasion of host immune responses, apoptosis resistance, and metastasis in various human cancers [26,36–39]. These findings suggest that drugs

targeting the JAK 2/STAT 3 pathway may hold great potential for a new novel therapeutic drug. This study observed hypophosphorylation in JAK 1 and JAK 2, suggesting that F8 inhibits this pathway reducing proliferation and angiogenesis and promoting apoptosis. As was the case for JAK 1 and JAK 2, their downstream effectors STAT 2, STAT 3, and STAT 5 were also found to be hypophosphorylated by F8 treatment. STATs represent a family of crucial transcription factors, driving the progression of cancer through hyperactivation or accumulated mutations that result in gain of function [40].

Excessive activation of STAT3 and STAT5 are implicated in numerous hematopoietic and solid malignancies, including chronic and acute myeloid leukemia, melanoma, and prostate cancer [40]. The heightened activity has been linked to upstream oncogenic triggers, such as the frequently mutated JAK 2 kinase [41]. Interestingly, hyperactive STAT 3 has been associated with large granular T-cell leukemia. Mutated STAT 5 has been identified in patients with T-cell prolymphocytic leukemia, T-ALL, T-cell-derived lymphoma, and monomorphic epitheliotropic intestinal T-cell lymphoma [42–44]. Here, we tested the compound F8 on the CCRF-CEM cell line and found reduced STAT 3 and STAT 5 activation. This reduced activation could be implicated in reducing proliferation and may directly induce apoptosis in the CCRF-CEM cell line.

In the near future, the effects of F8 on the phosphorylation of key signaling pathways will be extended to delineate further the critical molecules involved in cell death induction.

## Acknowledgments

The authors want to recognize Ms. Gladys Almodovar for the cell culture core. We also thank the Cellular Characterization and Biorepository Core Facility personnel at the Border Biomedical Research Center (BBRC) at the University of Texas at El Paso (UTEP) for their guidance and knowledge.

## Author Contributions

**Conceptualization:** Risa Mia Swain, Renato J. Aguilera.

**Data curation:** Risa Mia Swain, Anahi Sanchez, Denisse A. Gutierrez, Armando Varela-Ramirez.

**Formal analysis:** Risa Mia Swain.

**Funding acquisition:** Renato J. Aguilera.

**Investigation:** Risa Mia Swain, Denisse A. Gutierrez.

**Methodology:** Anahi Sanchez, Denisse A. Gutierrez, Armando Varela-Ramirez.

**Validation:** Denisse A. Gutierrez.

**Writing – original draft:** Risa Mia Swain.

**Writing – review & editing:** Armando Varela-Ramirez, Renato J. Aguilera.

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
