## [Decision Letter · Decision Letter 0]

3 Sep 2023

PONE-D-23-24580Thiophene derivative induces cytotoxicity via the intrinsic apoptotic pathway on human acute lymphoblastic leukemia cellsPLOS ONE

Dear Dr. Aguilera,

Thank you for submitting your manuscript to PLOS ONE. After careful consideration, we feel that it has merit but does not fully meet PLOS ONE’s publication criteria as it currently stands. Therefore, we invite you to submit a revised version of the manuscript that addresses the points raised during the review process.

We look forward to receiving your revised manuscript.

Kind regards,

Wagdy Mohamed Eldehna, Ph.d

Academic Editor

PLOS ONE

Journal Requirements:

https://link.springer.com/article/10.1007/s10637-022-01266-y?code=fe9adc41-9b47-40f2-b321-c6ce7bb093af&error=cookies_not_supported

https://www.dovepress.com/getfile.php?fileID=78603

In your revision ensure you cite all your sources (including your own works), and quote or rephrase any duplicated text outside the methods section. Further consideration is dependent on these concerns being addressed.

"This work was partially supported by National Institute for General Medical Sciences grants SC3GM088069-03 and 5R16GM149379-02 to RJA.  In addition, partial support was provided by the National Institute of Minority Health and Health Disparities Research Centers at Minority Institutions grant 3U54MD007592). We also thank the Cellular Characterization and Biorepository Core Facility personnel at the Border Biomedical Research Center at the University of Texas at El Paso for their assistance in experimental procedures. "

"RJA received funding from National Institute for General Medical Sciences grants SC3GM088069-03 and 5R16GM149379-02.  

6. Please upload a new copy of Figures 6 and 7 as the detail is not clear. Please follow the link for more information: " ext-link-type="uri" xlink:type="simple">https://blogs.plos.org/plos/2019/06/looking-good-tips-for-creating-your-plos-figures-graphics/"
" ext-link-type="uri" xlink:type="simple">https://blogs.plos.org/plos/2019/06/looking-good-tips-for-creating-your-plos-figures-graphics/"

Reviewers' comments:

Reviewer's Responses to Questions

**Comments to the Author**

1. Is the manuscript technically sound, and do the data support the conclusions?

Reviewer #1: Partly

Reviewer #2: Partly

2. Has the statistical analysis been performed appropriately and rigorously? 

Reviewer #1: N/A

Reviewer #2: Yes

3. Have the authors made all data underlying the findings in their manuscript fully available?

Reviewer #1: No

Reviewer #2: No

4. Is the manuscript presented in an intelligible fashion and written in standard English?

Reviewer #1: Yes

Reviewer #2: Yes

5. Review Comments to the Author

Reviewer #1: The manuscript requires revision in order to improve the clarity and coherence of the abstract. Additionally, there is a need to clearly state the aim of the study. Furthermore, grammatical errors should be addressed, particularly in relation to the use of future tense instead of past simple tense.

Compound F8 was recently published as a preclinical candidate for the treatment of undifferentiated gastric cancer. However, the synthesis of compound F8 described in this paper does not correspond to the chemical name you provided. Therefore, we kindly request that you provide us with the purchasing sheet or donation document from ChemBridge DIVERset library for compound F8, including information on its structure, purity, and quantity ordered.

We would also appreciate clarification on what is meant by "said library set" in this context. Did you mean the above-mentioned library?

Additionally, we have a question regarding the dilution protocol used during your experiments for Compound F8, as well as for the positive control and compounds Paclitaxel and Cytochalasin-D.

Furthermore, we would like to inquire about the detection of cell death via the differential nuclear staining (DNS) assay. Specifically, we would like to know if the results were obtained after 48 hours or 96 hours of incubation. It would be helpful if you could attach the raw data from this experiment.

Lastly, we are interested in knowing if there were any in vivo experiments conducted to support your findings.

Moreover, it appears that all figures were not included in the manuscript.

Thank you for your attention to these matters.

Reviewer #2: In this study Swain et al designed and synthesized novel Thiophene derivative induces cytotoxicity via the intrinsic apoptotic pathway on human acute lymphoblastic leukemia cells. This work seems interesting, but there are still some important concerns that need to be addressed.

In introduction

1. Authors should provide rational for molecular design linking the main scaffold with the tested molecular target, “MAPK and JAK-STAT” and link to apoptosis induction. A figure should include previously reported compounds with the same moiety. What are the main regions for both proteins that should be targeted?

2. Authors should highlight the novelty of the included work.

In Results

3. Is this compound (F8) novel, or previously reported? If it’s novel, authors should illustrate the synthesis scheme, characterization, and authors should clarify the choice of this compound, specifically for the study?, only one compound for the study may not enough.

4. In the cytotoxicity table, authors should add the IC50 values as “Mean±SD” in one column.

5. What about application of the In silico approaches regarding Docking and ADME pharmacokinetics?

6. Please, indicate which method was used to calculate the CC50 value for cytotoxicity.

7. Regarding DNA fragmentation, authors should provide the original Gel image.

General issues

8. Figures should be introduced in a clearer way with better resolution.

9. Add full definition for all figures captions. All figure legends should be self-explanatory (e.g, the Ic50 value, incubation duration)

10. Please, double check on the plagiarism ratio, grammar issues and typo mistakes.

6. PLOS authors have the option to publish the peer review history of their article (what does this mean?). If published, this will include your full peer review and any attached files.

Reviewer #1: **Yes: **Mahmoud Rashed

Reviewer #2: No

---

## [Author Response · Author response to Decision Letter 0]

2 Nov 2023

Reviewer #1: 

1. The manuscript requires revision in order to improve the clarity and coherence of the abstract. Additionally, there is a need to clearly state the aim of the study. Furthermore, grammatical errors should be addressed, particularly in relation to the use of future tense instead of past simple tense.

Authors response:

Thank you for pointing this out -we neglected to state the aim from the beginning and did not mention this until the introduction. We have added the following to the beginning of the abstract (lines 23-28) and now feel that it flows much better: In an effort to identify novel anti-cancer agents, we employed a well-established High Throughput Screening (HTS) assay to assess the cytotoxic effect of compounds within the ChemBridge Diverset Library on a lymphoma cell line. This screen revealed a novel thiophene, F8 (methyl 5-[(dimethylamino)carbonyl]-4-methyl-2-[(3-phenyl-2-propynoyl) amino]-3-thiophenecarboxylate), that displays anti-cancer activity on lymphoma, leukemia, and other cancer cell lines

The overall aim of this study is to investigate the compound F8 as a potential anti-cancer agent. Several assays were conducted to gauge the cytotoxicity of the compound (phosphatidylserine externalization, reactive oxygen species generation, etc). It is stated in the abstract on lines 25-28, "This screen revealed a novel thiophene, F8 (methyl 5-[(dimethylamino)carbonyl]-4-methyl-2-[(3-phenyl-2-propynoyl) amino]-3-thiophenecarboxylate), that displays anti-cancer activity on lymphoma, leukemia, and other cancer cell lines.” Lines 38-39 state the parameters of our study: "These findings demonstrate that F8 has promising anti-cancer activity."

2. Compound F8 was recently published as a preclinical candidate for the treatment of undifferentiated gastric cancer. However, the synthesis of compound F8 described in this paper does not correspond to the chemical name you provided. Therefore, we kindly request that you provide us with the purchasing sheet or donation document from ChemBridge DIVERset library for compound F8, including information on its structure, purity, and quantity ordered.

Authors response:

In the undifferentiated gastric cancer study, they use a compound with the same name (F8) but our compound has a completely different structure and not at all related. Please refer to the image below and the link to the paper, Development of o-aminobenzamide salt derivatives for improving water solubility and anti-undifferentiated gastric cancer - PMC (nih.gov). The PMID for that paper is 37497111.

The structure of our F8 compound is shown below and can be found at https://www.hit2lead.com/result.asp?search=93458698. Also, please see attached LCMS data of the compound F8. In regards to the purity, it is stated on the Chembridge website, “EXPRESS-Pick™ Stock compounds (7-digit ID compounds): Minimum purity of 90% and identity confirmed using 1H-NMR and/or LC-MS/ELSD.” 

3. We would also appreciate clarification on what is meant by "said library set" in this context. Did you mean the above-mentioned library?

Authors response:

As mentioned in the manuscript, "said library set" refers to the ChemBridge DIVERset library where the compound was originally detected. 

4. Additionally, we have a question regarding the dilution protocol used during your experiments for Compound F8, as well as for the positive control and compounds Paclitaxel and Cytochalasin-D.

Authors response:

The dilution protocol is stated in the manuscript. In lines 95-99, "The experimental compound was obtained from the ChemBridge DIVERset library and was pre-diluted to 10 mM in dimethyl sulfoxide (DMSO). The compound (methyl 5-[(dimethylamino)carbonyl]-4-methyl-2-[(3-phenyl-2-propynoyl) amino]-3-thiophenecarboxylate), also known as F8, was found in said library set and used for the experiments detailed below. Aliquots and other concentrations of F8 were thawed and diluted with DMSO. The positive control, H2O2, was alternatively diluted in 1 x phosphate-buffered saline (PBS). Compounds Paclitaxel and Cytochalasin-D were also prepared in DMSO." 

5. Furthermore, we would like to inquire about the detection of cell death via the differential nuclear staining (DNS) assay. Specifically, we would like to know if the results were obtained after 48 hours or 96 hours of incubation. It would be helpful if you could attach the raw data from this experiment.

Authors response:

The cell line of interest was plated on day 1 and left overnight (24 h) for the cells to stabilize. The following day, the cells were treated with a range of F8 concentrations and incubated for 24 h to determine the 24 h CC50. The CC50 that were listed in the Table are 48 h concentrations. Please refer to lines 261-262 in the manuscript, stating, "All grown cell lines were incubated at 48 h with F8 to obtain their CC50 value."

6. Lastly, we are interested in knowing if there were any in vivo experiments conducted to support your findings.

Authors response:

There were no in vivo experiments conducted in this study.

7. Moreover, it appears that all figures were not included in the manuscript.

Authors response:

The figure that was mentioned in the text as "data not shown" was a cell cycle using CC50 value obtained after 24 h of incubation. This data was not shown due to the extensive damage to the cells over a 72 h period. We could not properly analyze alterations in the cell cycle without reducing the concentration and that experiment did not fit in the parameters of our study. The sentence indicating data not shown was removed (lines 333-335), as this information does not add or detract from the study. 

Reviewer #2: 

In this study Swain et al designed and synthesized novel Thiophene derivative induces cytotoxicity via the intrinsic apoptotic pathway on human acute lymphoblastic leukemia cells. This work seems interesting, but there are still some important concerns that need to be addressed.

In introduction

1. Authors should provide rational for molecular design linking the main scaffold with the tested molecular target, "MAPK and JAK-STAT" and link to apoptosis induction. A figure should include previously reported compounds with the same moiety. What are the main regions for both proteins that should be targeted?

Authors response:

Please refer to lines 424-426 in the discussion, stating, “This study observed hypophosphorylation in JAK 1 and JAK 2, suggesting that F8 inhibits this pathway, reducing proliferation and angiogenesis and promoting apoptosis." Although we hypothesize a possible role for this pathway, we have no proof for this assertion.

We have attempted many docking experiments in the past with various compounds and found the results unreliable. In other words, we have not found a direct link between a good docking score and actual inhibition of activity. Due to these negative experiences, we do not believe that adding docking experiments will contribute reliable data to the paper and have decided to omit such data. Moreover, at this time, we do not know the principal target enzyme inhibited by the F8 compound and therefore we do not have a target for docking. Once we have more data pointing to a particular pathway or molecule, we will definitely attempt docking experiments.

2. Authors should highlight the novelty of the included work.

In Results?

Authors response:

As mentioned earlier, we neglected to add important information about the main Aim of the project in the Abstract and the Introduction and have amended the MS to reflect this. We have added the following to the beginning of the Abstract (lines 23-28) and now feel that it flows much better: “In an effort to identify novel anti-cancer agents, we employed a well-established High Throughput Screening (HTS) assay to assess the cytotoxic effect of compounds within the ChemBridge Diverset Library on a lymphoma cell line. This screen revealed a novel thiophene, F8 (methyl 5-[(dimethylamino)carbonyl]-4-methyl-2-[(3-phenyl-2-propynoyl) amino]-3-thiophenecarboxylate), that displays anti-cancer activity on lymphoma, leukemia, and other cancer cell lines.”

We have also added the following in the Introduction (lines 51-53):” After screening 1300 compounds, a novel thiophene compound was found to elicit potent cytotoxicity on the Acute Lymphoblastic Leukemia (ALL) CCRF-CEM cell line using the Differential Nuclear Staining (DNS) assay [9].”

3. Is this compound (F8) novel, or previously reported? If it's novel, authors should illustrate the synthesis scheme, characterization, and authors should clarify the choice of this compound, specifically for the study?, only one compound for the study may not enough.

Authors response:

Upon conducting searches of the compound on Pubmed, SciFinder and ChemSpider, we did not find any publications or patents describing the F8 compound, thus indicating it is a novel compound.

4. In the cytotoxicity table, authors should add the IC50 values as "Mean±SD" in one column.

Authors response:

Please refer to line 274 where "Mean±S.D." has been added to Table 1. 

5. What about application of the In silico approaches regarding Docking and ADME pharmacokinetics?

Authors response:

The application of in silico docking or ADME pharmacokinetics did not fit in the parameters of our study. We wanted to investigate the compound's potential anti-cancer activity but it is a good suggestion to take into consideration for future studies. 

6. Please, indicate which method was used to calculate the CC50 value for cytotoxicity.

Authors response:

It is stated in the manuscript: The cytotoxic concentrations were determined as previously described [9] using a linear interpolator calculator available at (https://www.johndcook.com/interpolator.html). Please refer to lines 125-127.

7. Regarding DNA fragmentation, authors should provide the original Gel image.

Authors response:

Regarding DNA fragmentation, this was conducted by flow cytometry utilizing an assay that measures total DNA content per individual cell. This type of analysis was used for its ability to detect an apoptosis-associated DNA fragmentation pattern at the single-cell level, as manifested by an increase in the sub-G0/G1 cell subpopulation (hypodiploid) during the cell cycle analysis profile. Thus, the DNA fragmentation was not detected on a gel.

General issues

8. Figures should be introduced in a clearer way with better resolution.

Authors response:

 The resolution of Figures 6 and 7 has been improved.

9. Add full definition for all figures captions. All figure legends should be self-explanatory (e.g, the Ic50 value, incubation duration). 

Authors response:

Please refer to lines 291-294, 304-307, 315-317, 328-330, 343-346, and 374-377 for all the legend figures. Please refer to the example below, highlighting the concentration value and incubation duration. 

"Fig 2. F8's induction of apoptosis. F8 compound-induced apoptosis in CEM cells. F8 ability to induce apoptosis was assessed in a phosphatidylserine externalization assay and measured through flow cytometry. Analysis was performed following a 24 h incubation with 24 h CC50 and 2X CC50 (2.89 µM and 5.78 µM). Controls included DMSO as the vehicle, H2O2 as the positive, and untreated. Significant phosphatidylserine externalization was evident for the CC50 and 2X CC50, given the p-value p0.00001 (***)." 

10. Please, double check on the plagiarism ratio, grammar issues and typo mistakes.

Authors response:

Please note that significant changes were made in the manuscript in order to avoid plagiarism. Please refer to the document titled “Track changes” to view them. It is also important to take note that similar sentences marked as plagiarism were due to things we cannot change such as our institutional affiliation in addition several sentences in the Materials and Methods are due to techniques we have developed or routinely use in our laboratory and have added the appropriate references of our work.

---

## [Decision Letter · Decision Letter 1]

22 Nov 2023

Thiophene derivative induces cytotoxicity via the intrinsic apoptotic pathway on human acute lymphoblastic leukemia cells

PONE-D-23-24580R1

Dear Dr. Aguilera,

We’re pleased to inform you that your manuscript has been judged scientifically suitable for publication and will be formally accepted for publication once it meets all outstanding technical requirements.

Kind regards,

Wagdy Mohamed Eldehna, Ph.d

Academic Editor

PLOS ONE

Additional Editor Comments (optional):

Reviewers' comments:

Reviewer's Responses to Questions

**Comments to the Author**

1. If the authors have adequately addressed your comments raised in a previous round of review and you feel that this manuscript is now acceptable for publication, you may indicate that here to bypass the “Comments to the Author” section, enter your conflict of interest statement in the “Confidential to Editor” section, and submit your "Accept" recommendation.

Reviewer #1: All comments have been addressed

Reviewer #2: All comments have been addressed

2. Is the manuscript technically sound, and do the data support the conclusions?

Reviewer #1: Yes

Reviewer #2: Yes

3. Has the statistical analysis been performed appropriately and rigorously? 

Reviewer #1: Yes

Reviewer #2: Yes

4. Have the authors made all data underlying the findings in their manuscript fully available?

Reviewer #1: Yes

Reviewer #2: Yes

5. Is the manuscript presented in an intelligible fashion and written in standard English?

Reviewer #1: Yes

Reviewer #2: Yes

6. Review Comments to the Author

Reviewer #1: The authors have addressed all the requirements, including all comments, figures, and answers to all questions, in a very satisfactory way.

Reviewer #2: Authors addressed all comments, and the manuscript quality was developed accordingly. Some additional comments, authors should check the grammar issues.

7. PLOS authors have the option to publish the peer review history of their article (what does this mean?). If published, this will include your full peer review and any attached files.

Reviewer #1: **Yes: **Mahmoud Rashed

Reviewer #2: No

---

## [Editor Report · Acceptance letter]

5 Dec 2023

PONE-D-23-24580R1 

Thiophene derivative inflicts cytotoxicity *via* an intrinsic apoptotic pathway on human acute lymphoblastic leukemia cells 

Dear Dr. Aguilera:

I'm pleased to inform you that your manuscript has been deemed suitable for publication in PLOS ONE. Congratulations! Your manuscript is now with our production department. 

Kind regards, 

on behalf of

Dr. Wagdy Mohamed Eldehna 

Academic Editor

PLOS ONE